# Permafrost Degradation Leads to Biomass and Species Richness Decreases on the Northeastern Qinghai-Tibet Plateau

**DOI:** 10.3390/plants9111453

**Published:** 2020-10-28

**Authors:** Xiaoying Jin, Huijun Jin, Xiaodong Wu, Dongliang Luo, Sheng Yu, Xiaoying Li, Ruixia He, Qingfeng Wang, Johannes M. H. Knops

**Affiliations:** 1Northeast-China Observatory and Research-Station of Permafrost Geo-Environment-Ministry of Education, Institute of Cold-Regions Science and Engineering, School of Civil Engineering, Northeast Forestry University, Harbin 150040, China; jinxiaoying@lzb.ac.cn (X.J.); luodongliang@lzb.ac.cn (D.L.); Sheng@lzb.ac.cn (S.Y.); lixiaoying@lzb.ac.cn (X.L.); ruixiahe@lzb.ac.cn (R.H.); qf_w@lzb.ac.cn (Q.W.); 2State Key Laboratory of Frozen Soils Engineering, Northwest Institute of Eco-Environment and Resources, Chinese Academy of Sciences, Lanzhou 730000, China; 3Cryosphere Research Station on the Qinghai-Tibetan Plateau, State Key Laboratory of Cryospheric Science, Northwest Institute of Eco-Environment and Resources, Chinese Academy of Sciences, Lanzhou 730000, China; wxd565@163.com; 4School of Forestry, Northeast Forestry University, Harbin 150040, China; 5Department of Health and Environmental Sciences, Xi’an Jiaotong Liverpool University, Suzhou 215123, China; Johannes.knops@xsjtlu.edu.cn

**Keywords:** active layer thickness, alpine vegetation, permafrost degradation, climate warming, soil water content

## Abstract

Degradation of permafrost with a thin overlying active layer can greatly affect vegetation via changes in the soil water and nutrient regimes within the active layer, while little is known about the presence or absence of such effects in areas with a deep active layer. Here, we selected the northeastern Qinghai-Tibet Plateau as the study area. We examined the vegetation communities and biomass along an active layer thickness (ALT) gradient from 0.6 to 3.5 m. Our results showed that plant cover, below-ground biomass, species richness, and relative sedge cover declined with the deepening active layer, while the evenness, and relative forb cover showed a contrary trend. The vegetation indices and the dissimilarity of vegetation composition exhibited significant changes when the ALT was greater than 2.0 m. The vegetation indices (plant cover, below-ground biomass, evenness index, relative forb cover and relative sedge cover) were closely associated with soil water content, soil pH, texture and nutrient content. Soil water content played a key role in the ALT–vegetation relationship, especially at depths of 30–40 cm. Our results suggest that when the ALT is greater than 2.0 m, the presence of underlying permafrost still benefits vegetation growth via maintaining adequate soil water contents at 30–40 cm depth. Furthermore, the degradation of permafrost may lead to declines of vegetation cover and below-ground biomass with a shift in vegetation species.

## 1. Introduction

The climate has been warming during the past decades, and a 0.3–4.8 °C increase in global mean annual air temperature has been projected for the 21st century [1]. The warming has led to substantial degradation of permafrost, as evidenced by deepening active layers (the layer above permafrost which thaws in summer and refreezes in winter) and rising ground temperatures [2,3,4,5,6]. The greatest warming of permafrost has been observed in the zones of the coldest permafrost; in the High Arctic at an average rate of 0.04 °C/yr (2007–2016) [2], and at a rate of 0.02 °C/yr (1980–2017) in the Interior of the Qinghai-Tibet Plateau (QTP) [7]. Meanwhile, data from field observations show a globally deepening active layer in permafrost regions [8,9]. In the continuous permafrost zone in the western Russian Arctic, the active layer thickness (ALT) in tundra increased at a rate of 0.62 cm/yr from 1997 to 2018 [10], while in the central areas on the QTP, the ALT increased at a rate of 1.95 cm/yr from 1981 to 2018 [11].

Permafrost degradation leads to vegetation changes, including changes in vegetation cover [12], species diversity [13], biomass [14], and plant species composition [15]. Vegetation changes can follow two pathways: (1) from terrestrial vegetation to aquatic vegetation coupled with forest, to sedges and mosses in lowland boreal forest underlain by ice-rich permafrost [16]. Permafrost degradation results in the melting of ground ice and increases the supra-permafrost water table, further causing the mortality of trees and the colonization of aquatic herbaceous plants [17]; (2) from hygrophilous to meso-xerophytic and xerophytic species [18]. For example, in the High Arctic, moss could be replaced by herbaceous plants [19,20], while on the QTP, alpine meadows are shifting to alpine steppes; meanwhile, a decline in vegetation cover and diversity has been observed [21,22]. Those vegetation processes are mainly attributed to changes in soil hydrology and the active layer [20,23]. It was found that permafrost degradation wets the soils in lowlands and dries the surface soils in uplands or on slopes [24] because of the different soil drainage in the active layer. In ice-rich lowlands at high latitudes, melting of ground ice and deepening of the active layer are associated with water-lodging conditions of soils in the active layer, because of the poor soil drainage and the relatively thin active layer [23]. However, on uplands, a deepening active layer, as a result of the melting ground ice, increases downward infiltration of water in soil [25] and decreases soil water content in the upper part [18]. From the interactions between thawing permafrost and soil water, the relationship between vegetation growth and ALT weakens with a thickening active layer. In addition to soil water content, soil nutrient content is also associated with ALT, because nutrient cycles, including accumulation, leaching, and utilization, are all associated with the dynamics of plants, microbes, and soil hydrology [26].

There have been several reports on the relationships between the ALT and vegetation in the Pan-Arctic regions. In comparison with that in arctic tundra, on average, the ALT is greater on the QTP due to its lower latitudes and stronger solar radiation [14]. Previous studies on the QTP have documented that increase in ALT reduces vegetation cover and above-ground biomass [27], with an ALT, or depth of the permafrost table, ranging from about 2 to 5 m; the greatest vegetation cover and biomass occurred in the areas with an ALT of about 2 m, and; vegetation cover, plant species diversity, and biomass decreased with the deepening active layer (from about 2 to 5 m) [21], coupled with vegetation changes from alpine meadows dominated by cold-wet adapted species to alpine steppes dominated by warm-dry species, and even to alpine deserts [5,27]. However, the ALT for triggering vegetation succession and for different succession trajectories remains poorly understood, and little is known about the main influencing factors for vegetation in permafrost regions on the QTP [28].

This study focuses on the ALT–vegetation relationship at high elevations (above 4289 m a. s. l.) with an ALT ranges of 0.6 to 3.5 m. Our hypotheses are: (1) Biomass, plant species diversity, and vegetation cover decrease with deepening active layer, and decrease progressively when the active layer is deeper than about 2.0 m; (2) Plant species composition is similar among sites with an ALT less than about 2.0 m; (3) Soil water content at vegetation root zone (30–40 cm) in the active layer, as well as seasonally frozen ground, is the main controlling factor for vegetation indices and plant species composition. The results will deepen our understanding of the relationship between plant community and the underlying permafrost under a warming climate.

## 2. Materials and Methods

### 2.1. Study Area

This study was conducted in the Headwater Area of the Yellow River (HAYR), the catchment area above Duoshixia, Madoi, Qinghai Province (about 29,000 km^2^ between 33.7067°–35.4833°N and 95.8847°–98.8233°E; 4,294 to 5236 m a. s. l.) on the northeastern QTP, in Southwest China (Figure 1). The climate is continental. Based on data from meteorological stations, the mean annual air temperature was lower at the CLP2 site (−4.5 °C, 2011–2016) at higher elevations in the eastern HAYR, higher at the MDX2 site (−3.6 °C, 2011, 2014–2016) in the western HAYR, and warmest at the KQ2 site (−3.3 °C, 2015 to 2016) in the central HAYR [29]. The annual precipitation in the HAYR declines northwestwards [30]. The data from the Madoi Meteorological Station (34.9167°N, 98.2167°E; 4272.3 m a. s. l.) showed a mean annual air temperature at −3.5 °C, and a multi-year average of annual precipitation at 323 mm during the period from 1953–2018 (precipitation data for the period of 1953 to 2017), with significantly (*p* < 0.05) increasing trends of 0.03 °C/yr and 0.93 mm/yr, respectively [31]. Most (>80%) precipitation falls during the growing season, generally from June to August; the winters, generally from November to the next March, are relatively dry [30]. Snow cover is unstable due to the meagre snowfall and strong winds in winter [32]. The average annual evaporation potential is about 1000 mm, and solar radiation ranges from 6000–6500 MJ·m^2^ [33].

A mosaic of continuous, discontinuous, sporadic, and isolated patches of permafrost and seasonally frozen ground occur in the HAYR on the northeast QTP [5]. More than 85% of the areal extent of the HAYR is underlain by permafrost, which is largely controlled by elevation, but it is also strongly influenced by topography, vegetation cover, lithology, and local soil drainage [25,34]. The lower limit of discontinuous permafrost is found at the elevation of 4420 m a. s. l., on the northern flank of the Bayan Har Mountains (near the YNG1 site) [29]. Permafrost is generally thinner than 100 m, and relatively warm (≥−1 °C) and dry [29]. As observed in boreholes, the thickest permafrost is 75 m, and the lowest mean annual ground temperature, measured at the depth of zero annual amplitudes of ground temperatures (generally at depths of 15–25 m) is −1.8 °C, according to the measurements in established boreholes in the study area [35].

The alpine plant ecosystem is generally simple, and the prevailing vegetation types include alpine meadows and alpine steppes, while the most abundant species are *Kobresia* spp., and *Carex* spp. in the high-cold and mesic meadow, and *Potentilla bifurca*, *Aster diplostephioides* (DC.) C.B. Clarke, *Polygonum sibiricum*, *Gueldenstaedtia verna*, and *Leontopodium* spp. in the xeric steppe [36]. Alpine meadows, meanwhile, are distributed on the shadowy slopes and the moist, flat valley bottoms, while there are alpine steppes, on the sunny slopes and in relatively dry areas. In addition, alpine meadow and steppe soils prevail in the HAYR [37].

### 2.2. Field Work and Laboratory Analyses

Vegetation surveys were conducted in August 2017. A total of 22 sites were selected according to the spatial distribution of different permafrost types, including continuous permafrost in the southern HAYR, discontinuous permafrost in the south-central HAYR around the Sisters Lakes (Gyaring and Ngöring Lakes), and sporadic permafrost in the western HAYR along an ALT gradient in the HAYR (Figure 1). At each site, three 50 × 50 cm quadrats [38] were randomly set up for sampling the relatively uniform vegetation, to avoid the intensive destruction in the vicinity of the boreholes, for measurements of active layer processes, ground temperature, and/or water table. Plant community cover and species cover were visually estimated by percentage in each 50 × 50 cm quadrat, and the above-ground biomass was measured by cutting the vegetation at the ground-surface level in an area of 25 × 25 cm in each quadrat, to avoid a destructive harvest. Below-ground biomass and soil samples were collected at depths of 0–10, 10–20, 20–30, and 30–40 cm with a 5-cm diameter hand-held soil auger. All the biomass was oven-dried to constant weight at 60 °C for 72 h. Soil samples were air-dried and sieved through a 2-mm sieve to remove roots and rock fragments >2 mm, and through a 0.05-mm sieve to obtain the content of silt and clay (named silt below because of the very small content of clay in the soil for this study area). Contents of soil total carbon and total nitrogen (TC and TN) were analyzed with a Maco-cube Elementar analyzer (Langenselbold, Germany), and the content of total phosphorous (TP) in the soil was measured by the sulfuric acid-perchloric acid digestion procedure [39]. Soil pH was measured in an 1:5 aqueous suspension using a pH meter. The ALT was obtained by interpolating ground temperatures measured in boreholes.

### 2.3. Data Analysis Methods

Plant community cover, above-ground biomass, plant species richness, and Pielou evenness index were used to describe the characteristics of vegetation. Plant species richness was counted based on the number of species, and Pielou evenness index was calculated. The Pielou evenness index is an indicator for the distributive evenness of an individual plant species in a vegetation community, which was based on the Shannon–Wiener index and positively correlates with species diversity [40]. Pielou evenness index (*J*) was calculated by using the following formula,
(1)$$J=\left(-\sum^{\;}P_{i}lnP_{i}\right)/ln\left(S\right)$$
where *Pi* = Ni/N. N is the total cover of all species in the quadrat; Ni is the cover of an individual species i; and *S* is the total number of plant species in each quadrat.

Functional groups were also classified for indicating the vegetation patterns. According to their growth form, plant species were divided into five functional groups: grasses, forbs, sedges, legumes, and shrubs in the study area [41]. Indicator species analysis was conducted to investigate the vegetation composition at the sites of permafrost and seasonally frozen ground using the indicspecies package [42].

Independent Student’s t tests or Mann–Whitney U tests (depending on the normality distribution of data) were used to identify the difference between the characteristics of vegetation and between soil properties at the same depth at study sites underlain by permafrost or seasonally frozen ground. Simple linear or curve fittings were used to examine the relationship between the ALT and the characteristics of vegetation, and between the ALT and soil properties (soil water content, pH, content of silt, TC, TN, and TP). Data normality and residual plot were checked before and after the linear or curve fittings. Pearson correlations were used to determine the relationships among the characteristics of vegetation and environmental factors. Step-wise multiple linear regressions of soil water content at different soil depths were used to identify the main depth of soil water content for affecting the vegetation indices. All abovementioned analyses were performed in SPSS software (Version 21). To identify the dissimilarity of plant composition, the method of non-metric multidimensional scaling (NMDS) [43] was applied by Bray–Curtis index [44]. Furthermore, envfit was used to identify the relationship between plant composition and the environmental variables [45]. The NMDS analysis and envfit plot were performed using the vegan package [46] in R [47].

## 3. Results

### 3.1. Characteristics of Environmental Factors in Areas of Permafrost and Seasonally Frozen Ground

Among the 22 study sites, 13 sites were located in the permafrost areas and 9 sites were in the area of seasonally frozen ground (Table 1). For permafrost sites, the ALT ranged from 0.6 to 3.5 m, with an average at 1.9 (±1.0) m (Table 1 and Table 2). The ALT is greater at lower elevations but smaller at higher elevations; so is the mean annual ground temperature. Soils at permafrost sites were dominated by fine particles, with an organic layer or peat under alpine meadows.

Soil water content was higher at sites with a shallow active layer, a thick surface organic layer, and high contents of silt, TC, TN, and TP, while other sites were characterized by a thick active layer, and lower silt and nutrient contents. At all frozen-ground sites, soil water contents ranged from 11% to 65% (volumetric), with an average at 35 (±18)%; pH was between 6 and 9, with an average at 8 (±1) (Table 2). In the study areas, silt content was low, varying from 0.2 to 5.2%, with an average at 2.0 (±1.6)%. The average contents of TC, TN, and TP were 48.4 (±29.4), 4.0 (±2.8), and 1.3 (±0.2) g/kg, respectively.

Compared with those in areas of seasonally frozen ground, soil water content TC, TN, and TP at all studied depths (0–40 cm) and silt contents at larger depths (20–40 cm) were higher at permafrost sites. However, soil pH was significantly higher at sites in the zone of seasonally frozen ground (Figure 2).

### 3.2. Characteristics of Vegetation in Areas of Permafrost and Seasonally Frozen Ground

In total, more than 46 vascular plant species were found at the 22 study sites, which were divided into five functional groups: sedges (2), forbs (38), grasses (3), legumes (1), and shrubs (1). The most abundant species were *Kobresia* sp., *Carex* sp., *Aster diplostephioides* (DC.) C.B. Clarke, *Polygonum sibiricum*, *Gueldenstaedtia verna*, *Leontopodium* sp., and *Potentilla bifurca* Linn. Indicator species analysis showed that *Carex* sp. was the indicator species (*p* < 0.001) of permafrost sites, while *Potentilla bifurca* Linn. was significantly (*p* < 0.01) associated with sites with seasonally frozen ground.

The average of community cover was 83 (±17)%. The values of above- and below-ground (at depths of 0–40 cm) biomass were 173.4 (±72.4) and 20791.4 (±9423.9) g/kg, respectively (Table 2). The average species richness was 6 (±2), and Pielou evenness of 0.7 (±0.1). The averages of relative cover of sedges and forbs were 53 (±33)% and 44 (±29)%, respectively.

In addition, plant community cover, belowground biomass, and sedge relative cover were higher at permafrost sites; and above-ground biomass, Pielou evenness, and relative forb cover were higher at sites in areas of seasonally frozen ground (Table 3). Belowground biomass, and relative cover of sedge and forb showed a significant difference between the two zonal types of the study sites (seasonally frozen ground and permafrost zones).

### 3.3. Distribution of Vegetation and Soil Properties along the ALT Gradient

At permafrost sites, plant cover (Figure 3A) and forb relative cover (Figure 3F) increased slightly at first and then decreased, while Pielou evenness index (Figure 3D) and relative sedge cover (Figure 3E) showed opposite trends, and below-ground biomass (Figure 3B) and speciess richness (Figure 3C) significantly decreased with deepening active layer. Pielou evenness index was influenced by the main functional groups of sedges and forbs. Plant cover, and relative cover of sedges decreased and forbs increased substantially at ALT greater than 2.0 m.

The non-metric multidimensional (NMDS) analysis showed the visualization of the distribution of vegetation sites based on plant species composition. Results showed more similar plant species composition at the ALT of 0.6–2.0 m, but a more scattered one at sites in the zone of seasonally frozen ground (Figure 4). This suggests more diverse distribution of plant species in the areas of seasonally frozen ground.

At permafrost sites, soil water content (Figure 5A), soil silt content (Figure 5C), TC (Figure 5D), TN (Figure 5E), and TP (Figure 5F) (with the exception of pH (Figure 5B)) showed significant decreasing trends with increasing ALT.

### 3.4. Relationship between Vegetation and Environmental Factors

The vegetation indices, except above-ground biomass and plant species richness, were significantly correlated (*p* < 0.05) with environmental factors (Table 4). Plant cover, below-ground biomass, and relative sedge cover were significantly positively (*p* < 0.05) correlated with soil water content, elevation, silt content, and soil nutrients (TC, TN, and TP), while Pielou evenness and forb relative forb cover were generally negatively correlated (*p* < 0.05) with these environmental factors. Correlation analysis showed that soil pH and contents of silt, TC, TN, and TP significantly (*p* < 0.05) increased with increasing soil water content (Table 5). NMDS analysis, with fitted vectors of the environmental factors, showed that vegetation species composition significantly correlated with environmental factors (Figure 6). Among all environmental factors, soil water content exhibited the highest influence on the vegetation in a linear fit (R^2^ = 62%, *p* < 0.001), followed by contents of soil TC (R^2^ = 57%) and TN (R^2^ = 53%). The individual explanation of site elevation and contents of silt and total phosphorous are all lower than 45%.

Furthermore, step-wise multiple linear regressions analysis (with soil water content at different depths as the predict variables) showed that all vegetation indices were associated with soil water contents at depths of 30–40 cm, except Pielou evenness (Table 6)

## 4. Discussion

Previous studies have focused on the relationship between permafrost and vegetation in permafrost areas with greater ALT (>1 m); and in areas of seasonally frozen ground, the final stage of permafrost degradation has not yet been closely examined. In this study, areas with ALT ranging from 0.6 to 3.5 m were investigated to represent areas at different stages of permafrost degradation. Furthermore, vegetation indices and soil properties in areas of seasonally frozen ground were also analyzed, to explore the vegetation succession trajectories and the main influencing factors during the process of permafrost degradation. The distribution of permafrost and ALT in the HAYR change with elevation, slope aspects, substrates, soil water content, and vegetation [48]. Thinner active layers are mainly distributed at higher elevations, with wetter, organically-richer, finer-grained soils, and with dense vegetation; while the seasonally frozen ground beneath alpine steppes is mainly distributed at lower elevations or on south-facing slopes, and soils are characterized by coarse sands and gravels, with good drainage conditions (Table 1). In addition, in areas of seasonally frozen ground, at the sites with an organic layer or dense vegetation, mean annual ground temperature is lower compared with other sites, indicating the importance of the vegetation and organic layer in modifying ground temperatures [49,50].

### 4.1. Changes in Vegetation Characteristics with the ALT

At the permafrost sites, vegetation indices significantly changed with the increase of ALT, suggesting that permafrost degradation could lead to vegetation degradation. With the increase of ALT, plant cover, below-ground biomass, species richness and relative sedges cover decreased, while relative forb cover increased. Our results consist with other studies on the QTP showing that vegetation cover and below-ground biomass decreased with the increased of ALT [22,27], and the dominant species changed from the Cyperaceae to mesoxerophytes and xerophytes [21]. Furthermore, our data also showed that vegetation indices changed dramatically when the ALT was more than 2.0 m, so as the vegetation composition, indicating that ALT was less than 2.0 m could maintain the alpine vegetation growth.

Indicator species analysis showed that *Carex* spp. was the indicator species for permafrost sites, while *Potentilla bifurca* Linn. was strongly correlated with sites at seasonally frozen ground. This is because in the zone of seasonally frozen ground, *Carex* L. was absent, but *Potentilla bifurca* Linn was common. This suggests that the decline of sedges over degrading permafrost may be mainly associated with the reduction of *Carex* L. Generally, at our sites, sedges prefer wetter and colder habitats with a shallower active layer, while forbs dominated in warmer and drier habitats with a thicker active layer. This pattern can be explained in that a shallower active layer is generally associated with higher soil water content or even with saturation in soil profile with low oxygen availability. Sedges dominated in these low-oxygen or anaerobic conditions with a great quality of aerenchyma, which could transport oxygen downward to the root system, hence improving nutrient accessibility in soil [51].

The NMDS analysis suggested that the vegetation composition was similar with an ALT is of less than 2.0 m and other vegetation indices like plant cover, relative cover of sedge and forb, and evenness would change dramatically with the ALT at 2.0–3.5 m. It has been documented that the vegetation cover and biomass were higher with the ALT at 2.0 m [21], while a depth of permafrost table of less than 3.0 m would result in a high vegetation cover (> 85%) [27]. In addition, a recent study showed a positive correlation between the below-ground biomass and the ALT in the upper Heihe River basin in the Qilian Mountains on the northeastern QTP [14], in contrast to the results of this study. This can probably be attributed to the regional differences in climate and vegetation cover between the two study areas. The multi-year average of annual precipitation is 400 mm in the Heihe River basin, while it is 323 mm in our study area in the HAYR. On the other hand, the vegetation cover ranged from 64–94% in the upper Heihe River basin, while from 47–100% in the HAYR. In arctic tundra, a deeper active layer or thaw depth is generally associated with more aquatic species [24,52,53,54]. This is attributed to a much shallower, and generally wetter, active layer and poor drainage, resulting from thawing ice-rich permafrost in the arctic [17,55].

The above-ground biomass and relative forb cover in our study did not show a significant relationship with the ALT. This can be explained that climate warming could also increase forb abundance [56]. Our sites, at lower elevations, could also be affected by grazing of large herbivores. Experimental studies indicate that grazing exclusion could increase the above-ground biomass, plant cover, and plant functional groups [57].

### 4.2. Influencing Factors of Vegetation in Response to Permafrost Degradation

The relationship between ALT and vegetation has been attributed to the influence of soil hydrology and nutrient content on vegetation in permafrost regions [14,16,53,58]. Soil water contents at shallow depths (0–40 cm) are significantly higher at permafrost sites, while silt contents at depths of 20–30 and 30–40 cm, and soil nutrient contents are significantly lower at sites in the zone of seasonally frozen ground (Figure 2). At permafrost sites, contents of soil water and major nutrients (TC, TN, and TP) significantly (*p* < 0.05) declined with increasing ALT (Figure 5), and they were much lower at sites with ALT greater than 2.0 m. This indicates an ALT of 2.0 m as the critical depth for maintaining the higher water content and nutrient availability for the upper part of soil layers (0–40 cm). In our case, sites with ALT less than 2.0 m were all located in flat areas with a thick organic layer (> 40 cm), and in alpine meadows with a vegetation cover greater than 90%. These observations about surface organic layer and vegetation explain the 2.0 m as the critical value of ALT in maintaining local soil hydrology for the underlying permafrost. Our results also showed that vegetation indices and species composition were strongly influenced by the physical properties (water content, pH, and silt content) and nutrient contents (TC, TN, and TP) of soils, as well as elevation (Figure 6).

The results of step-wise multiple linear regressions (factors were water content of soil at different shallow depths) showed that plant cover, below-ground biomass, and relative cover of forbs and sedges are significantly influenced by soil water content at depths of 30–40 cm (Table 6), revealing the importance of soil water content in the root zone of alpine vegetation on the QTP [55,59]. This suggests that even if ALT is greater than 2.0 m, the presence of permafrost may still benefit vegetation growth via the positive effects of higher soil water contents at 30–40 cm in soil depth. On the QTP, 86% of the sedge roots are distributed in the top 30 cm of soils [60]. With deepening active layer, soil water infiltrates downward, resulting in drier soil at shallow depths, and possibly even mortality of wetland plants [61]. Consequently, this may further result in a lower vegetation cover, species richness, and below-ground biomass; and furthermore, more shallow-rooted plants, for example, sedges would shift to deep-rooted plants, such as forbs [55,62], because of the better water uptake capacity of forbs [63].

Soil pH and contents of silt, TC, TN, and TP significantly (*p* < 0.05) increased with increasing soil water content (Table 5), suggesting the association of all those soil properties with soil water content. Taking into account the correlations between vegetation and soil properties, it is evident that soil water content is the most important influencing factor for alpine meadow and steppe vegetation [60,64]. This is in agreement with the results from other studies on the QTP [38,57,65].

In addition, vegetation is also strongly affected by elevation and microtopography (e.g., slope aspect) as well as other environmental factors [66,67]. There are associations of elevation with air temperature and precipitation, and slope aspect could influence soil drainage [68,69]. For example, in our study areas, the KQ2 and KQ4 sites were about a few hundred meters apart in the same basin. KQ4 is on a sunny slope (about 5°), while KQ2 is on the bottom of an intermontane basin. The vegetation at the KQ4 site is alpine desert with some drought-tolerant species dominated by *Artemisia sieversiana Ehrhart* ex Willd. and *Saussurea hieracioides* Hook. F. However, that at the KQ2 site is alpine meadow, dominated by *Kobresia* sp. and *Polygonum sibiricum* Maxim.

The distribution of vegetation and its influencing factors were identified in this study. We focused on the relationships between characteristics of vegetation and ALT at high elevations, above 4200 m a. s. l. However, due to the limited accessibility and the harsh field-work conditions for long-term monitoring, the total site numbers were limited to only 22 sites, and only 9 were in the arer of seasonally frozen ground. An implementation plan for long-term vegetation monitoring with more diverse sites, and the normality of the data for statistics in areas of permafrost and seasonally frozen ground, as well as talik zones, is required for better understanding of their relationships.

## 5. Conclusions

Our results showed that plant cover, below-ground biomass, plant species richness, and relative cover of sedges decreased with deepening active layer, while Pielou evenness and relative forb cover showed a contrary trend. All those abovementioned vegetation indices exhibited a significant decrease or increase at the ALT of 2.0 to 3.5 m. Furthermore, plant species composition was more similar at the ALT of less than 2.0 m, and with a thick organic layer and dense vegetation. In the areas of seasonally frozen ground, the dominant vegetation change from sedges to forbs and the *Carex* L. disappears totally. However, plant species were more diverse in the zone of seasonally frozen ground. The statistical analysis suggests that soil properties, including soil water content, pH, contents of silt (and TC, TN, and TP) in the active layer, and elevation, were responsible for changes in vegetation over degrading permafrost. Especially, soil water content at depths of 30–40 cm was the most important factor for vegetation succession. We concluded that an ALT greater than 2.0 m over permafrost could still maintain higher contents of soil water and nutrients for the vegetation root zone to sustain vegetation growth.

This study could help better understand the impact of increasing ALT on vegetation growth and composition, and allow a better prediction of vegetation succession above degrading permafrost. The impact of ALT on vegetation was studied, however, without accounting for changes in precipitation due to the limited meteorological stations or rain gauges in our study area. Because the crucial role of soil moisture content in influencing vegetation features is crucial, it is clear that the vegetation degradation is not only resulted from the thickening active layer; changes in precipitation also greatly impact alpine vegetation. Under a changing climate, especially those that are warmer and wetter climate, long-term monitoring and regional studies on the impacts of permafrost degradation, are urgently needed along with research on the dynamics of the supra-permafrost water table with alpine vegetation on the QTP, are urgently needed for scientific management, and prudent protection, of alpine ecosystems.

## Figures and Tables

**Figure 1 plants-09-01453-f001:**
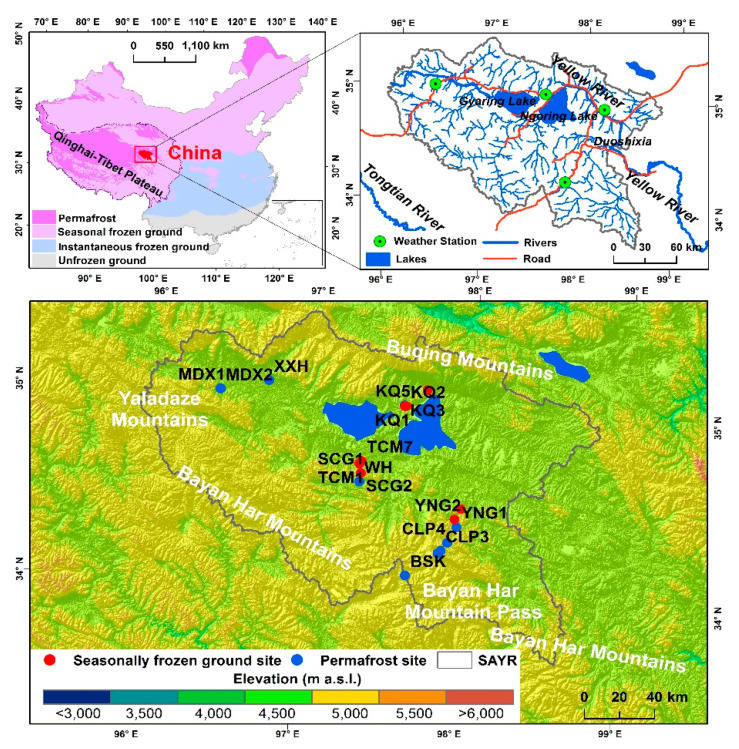
Study area and site locations in the Headwater Area of the Yellow River on the northeastern Qinghai-Tibet Plateau, Southwest China. All the sites are named based on name of the village nearby or the topography.

**Figure 2 plants-09-01453-f002:**
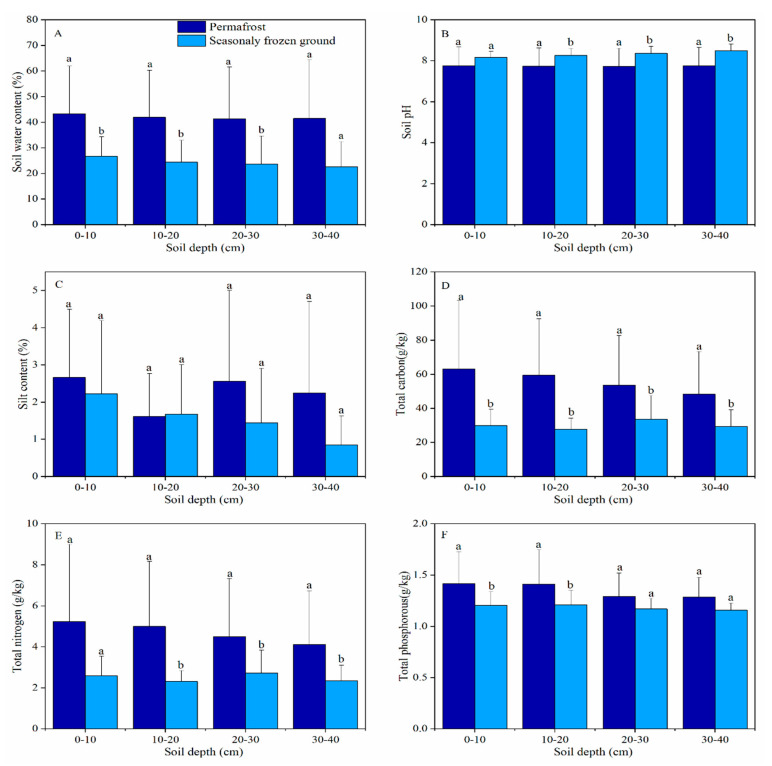
Soil physical properties and nutrient content in permafrost and seasonally frozen ground. Notes: (**A**), soil water content; (**B**), soil pH; (**C**), silt content; (**D**), soil total carbon; (**E**), soil total nitrogen; and (**F**), soil total phosphorous. N = 22. Different letters (a and b) above the bars means significant difference between sites at permafrost and seasonally frozen ground, using Student’s *t* tests or Mann–Whitney U tests.

**Figure 3 plants-09-01453-f003:**
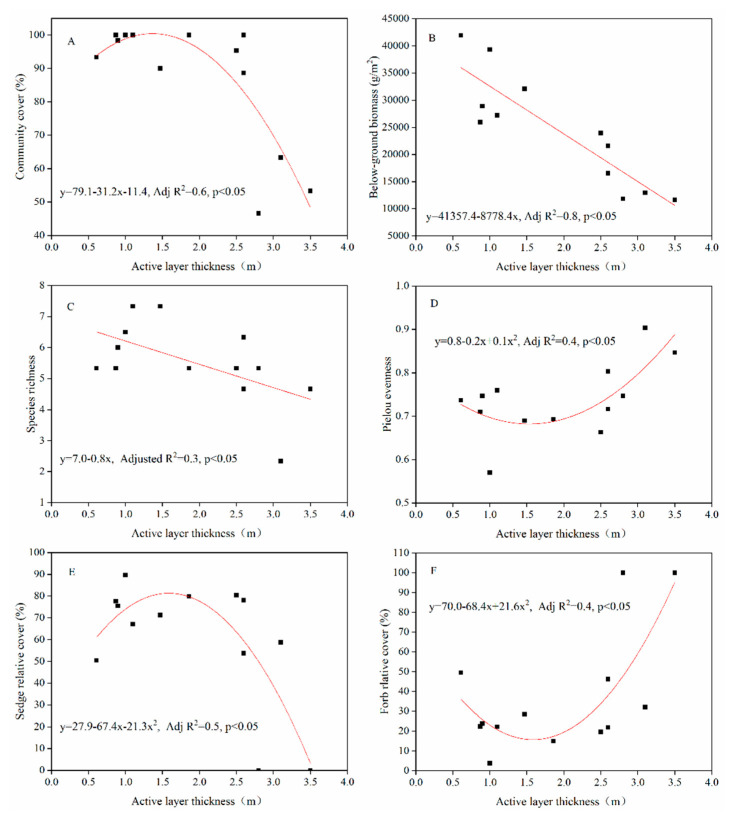
Relationships between vegetation indices and the active layer thickness in the permafrost sites. Notes: (**A**), community cover; (**B**), below-ground biomass; (**C**), species richness; (**D**), Pielou evenness; (**E**), sedge relative cover; and (**F**), forb relative cover. N = 13; data were averaged for the site level.

**Figure 4 plants-09-01453-f004:**
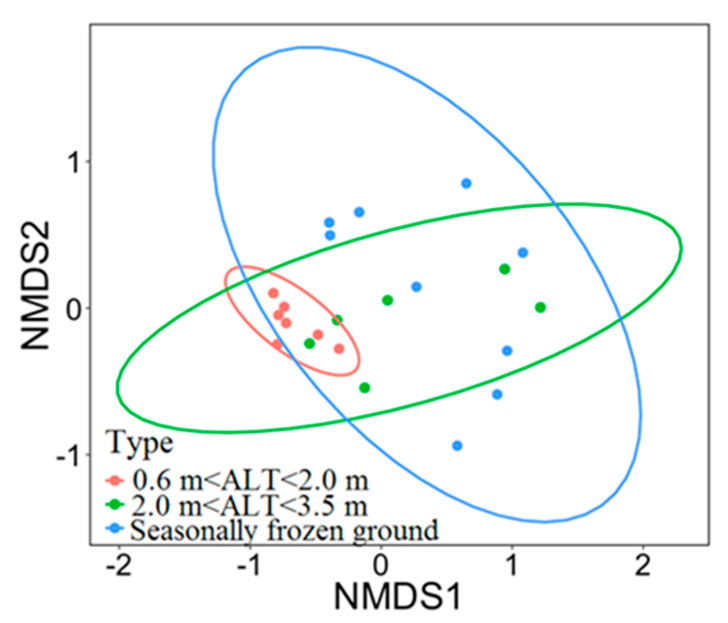
Non-metric multidimensional (NMDS) plot for the vegetation composition. The sites were divided into three categories based on their active layer thickness (ALT) at permafrost sites and at sites in areas of seasonally frozen ground (k = 2, stress = 0.15). Lines are standard dispersion ellipses, showing 95% confidence interval.

**Figure 5 plants-09-01453-f005:**
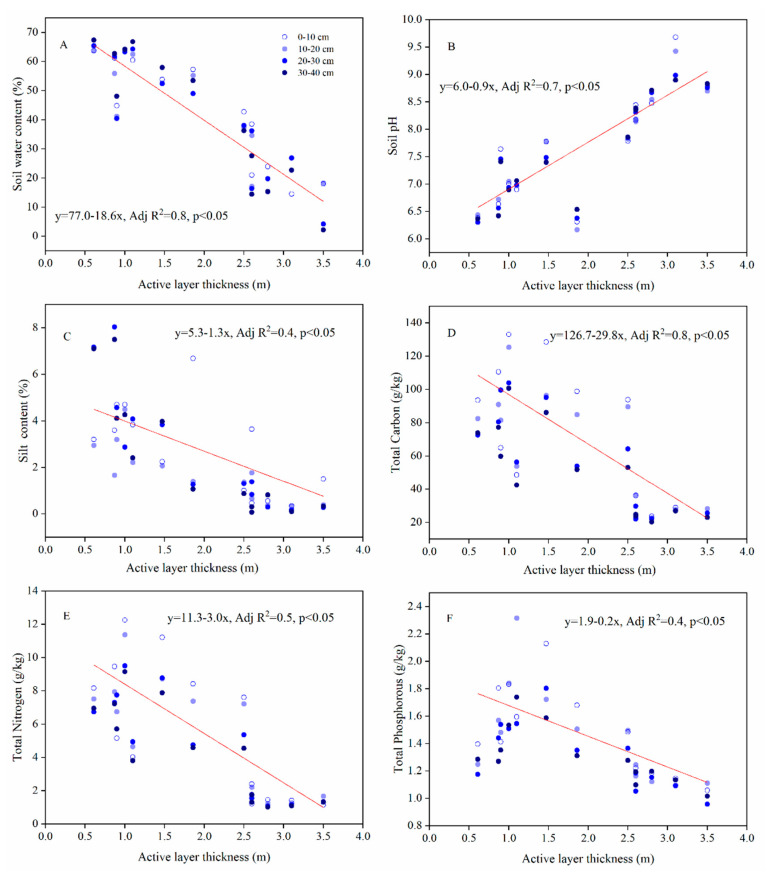
Relationships between soil properties and the active layer thickness. Notes: (**A**), soil water content; (**B**), soil pH; (**C**), silt content; (**D**), soil total carbon; (**E**), soil total nitrogen; and (**F**), soil total phosphorous.

**Figure 6 plants-09-01453-f006:**
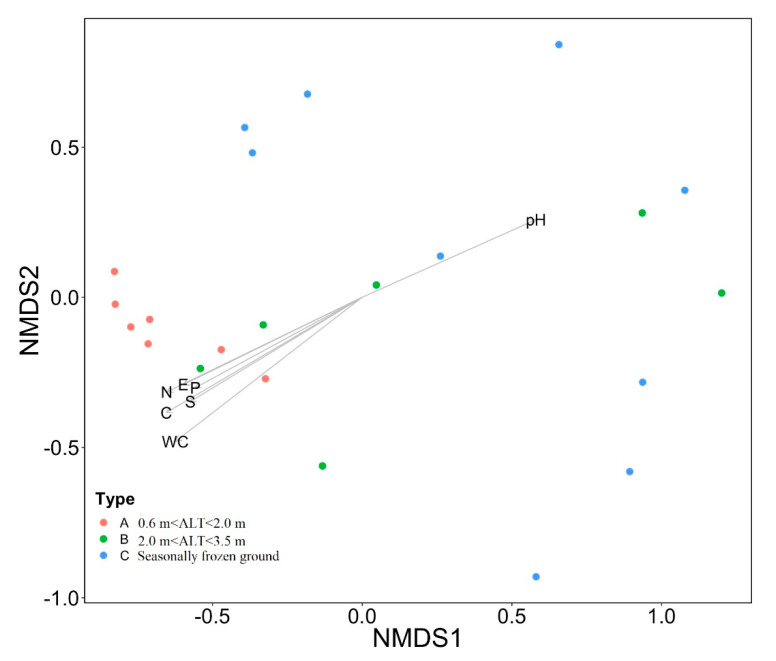
Non-metric multidimensional (NMDS) plot for the vegetation composition, with linear vectors of environmental factors (K = 2, Stress = 0.16). Note: Active layer thicknesses (ALTs) were grouped in three ranges: A, 0.6 m < ALT < 2.0 m; B, 2.0 m <ALT < 3.5 m; and C, seasonally frozen ground. Data were collected at 22 sites, with three replicates per site, and averaged for each site. The direction of the linear arrow refers to environmental gradient. Additionally, the length of vector represents the prediction of power. WC, soil water content; EC, electrical conductivity; E, Elevation; C, soil total carbon; N, soil total nitrogen; P, soil total phosphorous.

**Table 1 plants-09-01453-t001:** Site information in permafrost and seasonally frozen ground in the study area.

Sites	Longitude (°E)	Latitude (°N)	ALT (m)	MAGT (°C)	Elevation (m a. s. l.)	Slope (°)	Aspect	Topography	Soil Texture	Vegetation
BSK	97.6551	34.1287	0.61	−1.6	4833	5	west	Mountain pass	Fine sand	AM
CLP2	97.8496	34.2563	0.87	−1.7	4727	0	0	High flat plateau	Silt and loam	AM
TCM1	97.3141	34.6967	0.90	−0.2	4340	2	east-facing	Hill foot	Fine sand	AM
CLP4	97.9037	34.3148	1.00	−0.6	4564	0	0	In the valley	Fine sand	AM
SCG1	97.3269	34.6491	1.10	−0.4	4411	3	east-facing	Alluvial fan	Fine sand	AM
YNG1	97.9546	34.4013	1.47	−0.1	4452	4	east	Intramontane basin	Fine sand	AM
CLP3	97.8667	34.2706	1.86	−1.1	4663	0	0	mountain top	Fine sand with coarse gravels	AM
WL	97.3220	34.6040	2.50	N/A	4427	9	east	Lake bank	fine sand	AM
MDX1	96.3961	35.0367	2.60	N/A	4421	0	0	basin	Fine sand with gravels	AM
MDX2	96.3950	35.0350	2.60	N/A	4423	0	0	basin	Coarse sand with gravels	AM
KQ4	97.5708	35.0211	2.80	−0.6	4291	5	east	slope	Coarse sand with gravels	AD
KQ2	97.5734	35.0167	3.10	−0.4	4294	0	0	basin	Fine sand	AM
XXH	96.6976	35.0999	3.50	−0.2	4334	0	0	Alluvial fan	Coarse sand with gravels	AS
SCG2	97.3278	34.6491	N/A	N/A	4405	4	east	Alluvial fan	Fine sand with gravel	AS
TCM6	97.3248	34.7101	N/A	N/A	4311	0	0	basin	Fine sand with gravels	AS
TCM4	97.3124	34.6986	S	1.4	4326	1	south	basin	Coarse sand with gravels	AS
KQ1	97.5823	35.0175	S	1.0	4289	6	6/south	slope	Fine sand	AM
KQ3	97.5731	35.0166	S	0.4	4295	5	5/south	slope	Fine sand with gravels	AD
YNG3	97.9747	34.4962	S	1.1	4333	0	0	Plain	Fine sand	AM
YNG2	97.9396	34.4403	S	1.2	4395	9	9/south	slope	Coarse sand with gravels	AS
TCM7	97.3258	34.7127	S	0.4	4310	1	south	basin	Fine sand	AD
ELH	97.7081	35.1102	S	1.5	4365	0	0	Lake bank	Coarse sand with gravels	AS

Note: ALT, active layer thickness, and MAGT, mean annual ground temperature; S, seasonally frozen ground; AM, alpine meadow, AD, alpine desert, AS, alpine steppe, and; N/A, not available.

**Table 2 plants-09-01453-t002:** Soil environment and characteristics of vegetation at permafrost sites in the headwater area of the Yellow River on the northeastern Qinghai-Tibet Plateau, Southwest China. N = 13. Data at different depths were averaged for the site level.

Variables	Min	Max	Avg	SD
Active layer thickness (m)	0.6	3.5	1.92	1.0
Soil water content (%)	11%	65%	35%	18%
Elevation (m a. s. l.)	4289	4833	4419	149
Soil pH	6	9	8	1
Silt content (%)	0.2%	5.2%	2.0%	1.6%
Total carbon (g/kg)	19.7	115.8	48.4	29.4
Total nitrogen (g/kg)	1.2	10.6	4.0	2.8
Total phosphorous (g/kg)	1.1	1.8	1.3	0.2
Community cover (%)	47%	100%	83%	17%
Above-ground (g/m^2^)	64.3	325.7	173.4	72.4
Below-ground (g/m^2^)	1163.4	41,966.0	20,791.4	9423.9
Species richness	2	9	6	2
Pielou evenness index	0.6	1	0.7	0.1
Sedge relative cover (%)	4%	100%	53%	33%
Forb relative cover (%)	0	90%	44%	32%

**Table 3 plants-09-01453-t003:** Characteristics of vegetation in permafrost and seasonally frozen ground. N = 22, including sites in areas of permafrost and seasonally frozen ground.

Vegetation Indices	Permafrost (Avg ± SD)	Seasonally Frozen Ground	*p*
Community cover (%)	87 (±19)	77 (±13)	>0.05
Above-ground biomass (g/m^2^)	156 (±84)	199 (±43)	>0.05
Below-ground biomass (g/m^2^)	22,495 (±10,192)	14,443 (±1230)	<0.01
Species richness	6 (±1)	6(±2)	>0.05
Pielou Evenness	0.74 (±0.08)	0.81 (±0.05)	>0.05
Forb relative cover (%)	37 (±30)	75 (±23)	<0.05
Sedge relative cover (%)	60 (±29)	21 (±20)	<0.05

**Table 4 plants-09-01453-t004:** Significant correlations (*p* < 0.05) between vegetation and environmental factors. N = 22, including sites in areas of permafrost and seasonally frozen ground.

Environmental Factors	Community Cover (%)	Below-Ground Biomass (g/m^2^)	Pielou Evenness	Forb Relative Cover (%)	Sedge Relative Cover (%)
Soil water content (%)	0.7	0.9	−0.7	−0.6	0.6
Elevation (m a. l. s.)	0.5	0.8	−0.5	−0.5	0.5
Soil pH	−0.7	−0.9	0.6	0.6	−0.6
Silt content (%)	0.5	0.7	−0.5		0.5
Total carbon (g/kg)	0.6	0.9	−0.7	−0.7	0.7
Total nitrogen (g/kg)	0.6	0.9	−0.7	−0.6	0.6
Total phosphorous (g/kg)	0.7	0.7	−0.6	−0.7	0.6

Notes: Above-ground biomass and species richness do not show any correlations with any environmental variables. Hence, they are not presented in this table.

**Table 5 plants-09-01453-t005:** Correlation between environmental factors. N = 22, including sites in frozen ground areas.

	Soil Water Content (%)	Elevation (m a. s. l.)	pH	Silt Content (%)	Total Carbon (g/kg)	Total Nitrogen (g/kg)	Total Phosphorous (g/kg)
Soil water content (%)	1						
Elevation (m a. s. l.)	0.8 **	1					
pH	−0.9 **	−0.8 **	1				
Silt content (%)	0.8 **	0.7 **	−0.8 **	1			
Total carbon (g/kg)	0.9 **	0.7 **	−0.8 **	0.8 **	1		
Total nitrogen (g/kg)	0.9 **	0.7 **	−0.8 **	0.8 **	1 **	1	
Total phosphorous (g/kg)	0.9 **	0.4	−0.7 **	0.6 **	0.8 **	0.8 **	1

Note: ** *p* < 0.01.

**Table 6 plants-09-01453-t006:** Step-wise multiple linear regressions of soil water content from different depths (0–10 cm, 10–20 cm, 20–30 cm, 30–40 cm). N = 22, including sites in frozen ground areas.

Variables	Regression Models	R_adj_^2^	*p*
Community cover (%)	y = 64.1 + 55.9 WC (30–40 cm)	0.4	<0.01
Below-ground biomass (g/m^2^)	y = 7860.4 + 39149.0 WC (30–40 cm)	0.8	<0.01
Pielou evenness index	y = 0.9-0.4 WC (0–10 cm)	0.5	<0.01
Sedge relative cover (%)	y = 10.9 + 98.4 WC (30–40 cm)	0.4	<0.01
Forb relative cover (%)	y = 88.4–105.4 WC (30–40 cm)	0.4	<0.01

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
