# Peer review of "Permafrost Degradation Leads to Biomass and Species Richness Decreases on the Northeastern Qinghai-Tibet Plateau"

_plants, 2020, doi:10.3390/plants9111453_

Round 1

Reviewer 1 Report

The paper „Permafrost degradation leads to decrease in biomass and species richness on the northeastern Quinghai-Tibet Plateau“ researches the impact of the active layer thickness on biomass, vegetation composition, species richness and plant cover. It may contribute to a deeper understanding of the consequences of climate change on vegetation dynamics and succession trajectories. However, I think that within the paper statistical methods, results and discussion are poorly connected. Some methods are used but are not described in the material and methods section (stepwise regression). Moreover, the full potential of the multivariate NMDS ordination is not used (overlay arrows), instead multiple but simple testing is applied to each variable. Here I suggest to use multifactorial or multivariate models and testing in order to give a more holistic overview of the data. I also believe that a deeper analysis of the vegetation data including an indicator species analysis and a representation of the functional types would benefit the paper.
L27: significant changes
L28: Name the vegetation indices.
L29: Please correct the sentence structure
L 33: Please add a half sentence to decribe the ecological degradation.
L47: Either state the rate or the sum to make both observations comparable.
L52: Please rephrase this paragraph trajectorie 2) is missing.
L84: It remains unclear what you mean by site scale.
L85: Here I do nnot understand which part of your analysis answers the question raised.
L95:Based?
Figure 1: Please consider altering figure 1. Graphically: align the legend and sort it logically (all point features, line features etc.), alter the elevation range so that the range is larger, I do not detect any brown colour in the image so maybe the range refers to the whole digital elevation model. Change the scale bar in the lower part so that meaningful units are applied.
Logically: It is difficult to see the sampling points a subset of 4 smaller maps could be a solution.

L126: It is not clear to me what you mean by site, plot and quadrats. How many plots are in permafrost and does each site have both classess?
L133: Which scale / method did you use to determine species cover?
L171. Again it is not clear what you mean by site /plot.
L188: To make your results more meaningful please calculate an indicator species analysis for permafrost and seasonally forzen ground / or for your ALTdepth classess.
Figure 4:Please show the reationships between plant species compostion and environmental factors via biplot arrows in the graphic. Are all red plots found at the same site?
Figure 2: Please add the name of the statistical test etc. to the label. Why did you chose students t, consider a Anova / when the variance is inhomogeneous a Kruskall Wallis Anova with respective post hoch testing and corrections for multiple testing. A colour scale with gradual colours would be nice.
L268: areas were

Stepwise linear regression is missing in the methods chapter.
Discussion: you mention the influence of slope, aspect etc. on layer thickness. All variables affect vegetation as well, it is thus important to either use multifactorial testing or to controll for these effects. Otherwise it is difficult to argue that permafrost degradation leads to biodiversity loss.

Author Response

Dear Reviewer 1,

Thank you very much for your comments and your hard work in improving the manusript. 

Please see the attachment of our revision.

Best,

Authors

Reviewer 2 Report

This manuscript reported the vegetation communities (diveristy and composition) and biomass along with ALT gradient at permafrost and seasonally frozen sites at QTP. To cover a wide range of the gradient, the authors energetically conducted vegetation survey and soil core sampling in field. Because of the coverage of ALT gradient and the survey and sampling efforts, the obtained result is high-quality and the study design sounds good. However, I found some points to be modified or improved in the manuscript. Please carefully adress them listed below.

Major comments

The authors investigated the relationships not only between ALT and vegetation community, but also between ALT and soil properties. In fact, vegetation must however be directly affected by the soil properties. Therefore, the relationship between vegetation composition and the soil properties should be additionally analyzed using envfit or db-RDA etc. In addition, the effects of examined factors can be fitted to the NMDS plot.

L84 Why did the authors set up the hypothesis(2)? I could not find any related descriptions to lead this hypothesis in introduction section.

L85 As the authors mentioned in the hypothesis(3), drawing water content gradient in NMDS plot (e.g. using ordisurf function in R vegan package) would make readers easy to understand the relationship between water content associated with ALT and vegetation composition.

L209 The authors said some indeces of vegetation changed at 2.0-m ALT. However, the authors' interpletation is subjective. To show the evidence, curve fitting using quadratic or generalized additive model (i.e. spline curve fitting) can be used. This would help to provide an objective indicator for the threshold of the changing point. Following to the reanalysis, modify Figure 3 and 5.

Minor comments

L29 En-dash "–" should be used to indicate the range of values through the manuscript (many matches).

L81 Use comma to show the digit as "4,289 m" through the manuscript (e.g. L194).

Figure 1 Maybe, no need the legend title "Legend". Remove it. In addition, a period at the last of figure description should be also removed because it is not a sentence.

L138 This sentence is redundant because the same sentence occurred in L135–136. Simplify it.

Figure 2 The figure label "Seasonally frozen groud" is partially hidden by layering. Modify it.

Figure 3 The label of (f), italicize "Carex". In addition, axis for ALT should be set in the range of 0.5–3.5 m as in Figure 5.

Figure 4 Aspect ration of NMDS plot must not be changed. Correctly scale them.

Author Response

Dear Reviewer 2,

Thank you very much for your nice comments and suggustions. We revsed the manuscript, please see the attchment.

Thank you again.

Best regards,

Authors

Reviewer 3 Report

Thank You for the opportunity for reading the manuscript titled "Permafrost degradation leads to a decrease in biomass and species richness on the northeastern Qinghai-Tibet Plateau". Similar studies provide us with scientific base for responsible and qualified decision-making, facing recent changes in almost all habitats on Earth.

The paper has a huge potential for reaching target readers; however, more changes are needed before publication. There is a lack in methodological part starting with unclear design fo data collection that can bias the results through lack of necessary tests allowing us to use specific methods (eg, normality test, the relevance of using linear models, etc.). There are more typos or unclear facts in the text that have to be corrected before publishing. My comments and suggestions are summarized below, they are also included in the attached pdf file. I hope there will be helpful for the authors.

Annotation Summary of plants-921045-peer-review-v1_rev.pdf.

Note [page 1]: QTB, abbreviation is needed, since this is showed in the rest fo the text and ALT si also mentioned here…

Highlight [page 1]: Here, we selected the northeastern Qinghai-Tibet Plateau as study area.

Note [page 2]: All abbreviations should be explained at one place, there will be no need to look for them in various part of the text.

Highlight [page 2]: 0.02 ËšC/yr (1980-2017) in the Interior of the Qinghai-Tibet Plateau (QTP) [7].

Note [page 2]: Paper #19 is not primarily about high arctic, please cite proper papers not secondary citations, here I propose some recent and relevant papers:

Frost, G.V., Epstein H.E., Walker, D.A., Matyshak G., & Ermokhina, K. 2018. Seasonal and Long-Term Changes to Active-Layer Temperatures after Tall Shrubland Expansion and Succession in Arctic Tundra. Ecosystems. 21:507-520. . doi: 10.1007/s10021-017-0165-5.

Bhatt, U.S., Walker, D.A., Raynolds, M.K., Bieniek, P.A., Epstein, H.E., Comiso, J.C., Pinzon, J.E., Tucker, C.J., Steele, M.A., Ermold, W. and Zhang, J. 2017. Changing seasonality of Pan-Arctic tundra vegetation in relation to climatic variables. Environmental Research Letters. 12:055003. . doi: 1088/1748-9326/aa6b0b.

Bratsch, S.N., Epstein, H.E., Buchhorn, M., Walker, D.A. and Landes, H.A. 2017. Relationships between hyperspectral data and components of vegetation biomass in Low Arctic tundra communities at Ivotuk, Alaska. Environmental Research Letters. 12:025003.

Frost, G.V., Epstein, H.E., Walker, D.A., Matyshak, G. and Ermokhina, K. 2017. Seasonal and long-term changes to active-layer temperatures after tall shrubland expansion and succession in arctic tundra. Ecosystems. 16, 1296. . doi: 1007/s10021-017-0165-5.

Highlight [page 2]: For example, in high arctic, moss could be replaced by herbaceous plants [19], while on the QTP, alpine meadows are shifting to alpine steppes; meanwhile, a decline in vegetation cover and diversity has been observed [20,21].

Note [page 2]: In the arctic, changes are linked also with changes in active layer! See the papers I suggested above.

Highlight [page 2]: Those vegetation processes are mainly attributed to changes in soil hydrology [22].

Note [page 3]: I would prefer different format of coordinates that is more frequent in scientific studies, e.g. decimal degrees, not degrees and minutes

Highlight [page 3]: This study was conducted in the Headwater Area of the Yellow River (HAYR), the catchment area above Duoshixia, Madoi, Qinghai Province (about 29,000 km 2between 33°42.4ʹ–35°29.0ʹN and 95°53.5ʹ–98°49.4ʹE; 4,294 to 5,236 m a.

Note [page 3]: “Based” not “based”

Highlight [page 3]: based on data from the meteorological stations, the mean annual air temperature was lower at CLP2 site (−4.5ËšC, 2011-2016) at higher elevations in the eastern HAYR, higher at MDX2 site (−3.6ËšC, 2011, 2014-2016) in the western HAYR, and warmest at KQ2 site …

Highlight [page 4]: Study area and site locations in the Headwater Area of the Yellow River on the northeastern Qinghai-Tibet Plateau, Southwest China.

Note [page 4]: There is many papers from Himalayas focusing on plant communities and vegetation. I would concentrate more on descriptions of individual types and describing their different ecology and environmental requirements.

Eg Chang, D. (1981). The Vegetation Zonation of the Tibetan Plateau. Mountain Research and Development, 1(1), 29-48. doi:10.2307/3672945

Kürschner, H., Herzschuh, U. & Wagner, D. 2005. Phytosociological studies in the north-eastern Tibetan Plateau (NW China) — A first contribution to the subalpine scrub and alpine meadow vegetation. Bot. Jahrb. Syst. 126/3: 273-315.

Miller, D. 2005. The Tibetan Steppe. Book chapter
https://www.researchgate.net/publication/239587003_The_Tibetan_Steppe

And much more can be found.

Highlight [page 4]: The alpine plant ecosystem is generally simple and the prevailing vegetation types include alpine meadows and alpine steppes, and the most abundant species are Kobresia sp., Carex sp., Aster diplostephioides (DC.) C.B. Clarke, Polygonum sibiricum, Gueldenstaedtia verna, Leontopodium sp. and Potentilla bifurca.

Note [page 4]: In percentage? It should be mentioned here.

Highlight [page 4]: Plant community cover and species cover were visually estimated, and the above-ground biomass was obtained by cutting over the vegetation at the ground-surface level in an area of 25×25 cm in each quadrat to avoid a destructive harvest.

Note [page 4]: Was counted base on number of species and Pielou evens index.

Highlight [page 4]: Plant species richness was estimated as the number of species and Pielou evenness index was calculated.

Note [page 4]: What does the evenness say about habitat quality or ongoing changes. It would be great if this is explained here or bellow in the text, because there should be clear explanation why this index has been chosen and not another one. What about using alpha, beta or/and gamma diversity??

Highlight [page 4]: The Pielou evenness index is an indicator for the distributive evenness of an individual plant species in a vegetation community [38].

Note [page 4]: Overall map is missing. It is necessary to think that people from all over the world will read this paper and they should be familiar with geographical position as quick as possible.

Note [page 5]: and shrubs.

Note [page 5]: An unpaired t-test or paired were used? Was the normality tested before using T test?

Strikeout [page 5]: According to their growth form, plant species were divided into five functional groups: grasses, forbs, sedges, legumes and shrub plants [39].

Highlight [page 5]: Student’s t tests were used to identify the difference between the characteristics of vegetation and between soils properties in the same depth at study sites underlain by permafrost or seasonally frozen ground.

Note [page 5]: Which type? What was the presumption to use linear models? Was the data distribution tested as prerequisite for using lineament models? This should be stated here.

Highlight [page 5]: Linear models were used to examine the relationship between the ALT and characteristics of vegetation, and between the ALT and soil properties (soil water content, pH, content of silt, TC, TN and TP).

Note [page 5]: Is this your measurements or extrapolated data from literature? If data from literature, this belongs to methods, no need to be stated in the Results section.

Highlight [page 5]: The ALT is greater at lower elevations but smaller at higher elevations; so is the mean annual ground temperature (ranging from −1.7ËšC at elevation of 4,727 m a. s. l. to −0.1ËšC at 4,452 m a. s. l.).

Note [page 5]: Above, 5 PFT groups are mentioned, please harmonize this.

Highlight [page 5]: In total, more than 46 vascular plant species were found at the 22 study sites, which were divided into four functional groups: sedges (2), forbs (38), grasses (3) and legumes (1).

Note [page 5]: Was there only one KObresia and Carex species or more? Why species has not been determined to species level? Lot of information can be missing here. Knowledge about species (mainly dominant) is basic precursor for any kind of botanical or ecological research.

Highlight [page 5]: The most abundant species were Kobresia sp., Carex sp., Aster diplostephioides (DC.) C.B. Clarke, Polygonum sibiricum, Gueldenstaedtia verna, Leontopodium sp.

Note [page 6]: Results

Highlight [page 6]: Resutls showed more similar plant species composition at the ALT of 0.6-2.0 m, but more scattered one at sites in the zone of seasonally frozen ground (Figure 4).

Note [page 6]: It is logically, because seasonality affect variability in relief and differentiation of environmental factors shaping plant community more than permafrost, that in certain level unify ecological conditions.…

Taking into account this information I am wondering how 22 sites were chosen and what was the design of data collection? It is crucial factor that can affect whole results and create many biases that could explain methodological issues more than obtained differences. Please pay more attention describing design of selection 22 sites. If the sites were chosen randomly within 2 types (with permafrost and seasonal frozen) this should be stated. If not, potential bias should be discussed or study must be repeated.

Highlight [page 6]: This suggests more diverse distribution of plant species in the areas of seasonally frozen ground.

Note [page 7]: Please divide slope and aspect into 2 columns. Why in some cases aspect is not stated? It should be written everywhere the slope was not equal to 0.

Highlight [page 7]: Slope (Ëš) and Aspect

Note [page 8]: This visualization can be pretty tricky, I would keep only same colors from various sites together because only them were compared.
You can arrange same depths /colors beside each other and keep them in a column as they decrease with the depth - blue first, then light blues, pink and orange ones.

Note [page 9]: Only this one is significant, I would suggest to use stars instead of this, eg ** - significant at 0.01, * - significant at 0.05, no star - non significant.

But: In the Table 3, there is p >0.05, but in the pictures- a,b,c,d there is different symbol: p<0.05. It looks like a mistake or type somewhere..

Picture a) looks like there is linear dependence, it looks rather logarithmic, linearity must be tested first!! By proper test. Also some other pictures look questionable.

It is hard to say more about these results when I do not know if the statistics are correct…

Highlight [page 9]: >0.05 <0.01

Note [page 12]: What was the main reason explaining various vegetation types? Was it ecology of the stands or effect of climate change/permafrost change? First we need to understand ecology and dynamics of unchanged communities and then - based on these information - we need to explain potential consequences of permafrost changes. Based on the methods presented here, it is hard to say, what is the real reason of the observed differences. We need to avoid potential tautology or circle argumentation in the research.

Highlight [page 12]: At the permafrost sites, vegetation species were mainly Kobresia Willd and Carex L., as well as Leontopodium haplophylloides Hand.-Mazz. However, in the zone of seasonally frozen ground, Carex L.

Note [page 12]: I doubt this can be said based on NMDS results without any additional data. If it was not tested or there is no correlation based on real data included in the NMDS analyses, this would be only a speculation and thus it shouldn’t be written here.

Highlight [page 12]: The NMDS analysis suggests that the active layer thickening has no significant effect on vegetation composition until reaching an ALT of…

Note [page 13]: Not only, maybe these key book should be cited here:

Körner, C. 2003. Alpine plant life. Functional plant ecology of high mountain ecosystems. 2nd edition. Springer-Verlag, Berlin, Heidelberg, New York, 344 p.

Highlight [page 13]: In addition, vegetation is also strongly affected by elevation and microtopography (e.g., slope aspect) [63].

Author Response

Dear Reviewer 3,

Thank you very much for your comments and the suggestions. We also appreciate your sharing your knowlege with us. We revised the manuscript accordingly, and please see the attachment.

Thank you again, and we wish you all the best.

Kind regards,

Authors

Reviewer 4 Report

see Add. file

Author Response

Dear Reviewer 4,

Thank you very much for your comments. We really appreciate your valueing our work.  Please see the attachment.

Thank you again, and we wish you all the best.

Kind regards,

Authors

Round 2

Reviewer 1 Report

Dear authors,

i think the paper already benefitted from the review process. I still belive that minor english editing might be neccessary. Also the map could be graphically improved (pink colours in a map are not wrong but look peculiar) by considering general readability (partly text is quite small).Please also check if all methods are described in the methods section.

Author Response

Dear reviewer 3,

Thank you very much for your comments and suggestions. We really appreciate your hard work.

We revised the manuscript according to your comments. Please see the attachment.

Best regards,

Authors

Reviewer 3 Report

Thank You for improving the previous version of the manuscript. Recently I have only a few suggestions and corrections that should be changed or explained before publishing. My recommendation is to publish a paper after minor revision that the authors can easily prepare in a short time. My review is summarized below and also written directly into the attached text.

Annotation Summary of plants-921045-peer-review-v2rev.pdf.

Note [page 1]: Is it really community cover and not “plant cover”? There are differences, community cover means cover of selected communities that have changed over the permafrost layer and plant(species) cover means cover of various plants are changing due to permafrost changes… Please be exact and do not change the meaning of community /vegetation) and plants (species/taxa).

Highlight [page 1]: The vegetation indices (community cover, below-ground biomass, evenness index, forb relative cover and sedge relative cover) were closely associated with soil water content, soil pH, texture and nutrient content.

Note [page 3]: There should be an explanation what individual location’s abbreviation mean in the Figure 1’s header. All pictures should be self-supporting. At lease reference to the liftoff localities should be added here thus readers can find meanings immediately.

Highlight [page 3]: Study area and site locations in the Headwater Area of the Yellow River on the northeastern Qinghai-Tibet Plateau, Southwest China.

Note [page 4]: If more species of one genus is in the mind, abbreviation “spp.” or “sp. div” (=species diversae) should be used. “Sp.” means only one particular species not determined to species level is taken into account. As you explained in your cover letter, you meant more species of Kobresia as well as Carex.

Highlight [page 4]: The alpine plant ecosystem is generally simple and the prevailing vegetation types include alpine meadows and alpine steppes, and the most abundant species are Kobresia sp., Carex sp.

Note [page 5]: Why except? Maybe I overlooked it, but if not, it should be explained here.

Note [page 5]: Same as my comment before.

Note [page 5]: Kobresia spp., Carex spp. - please correct everywhere in the text.

Highlight [page 5]: The most abundant species were Kobresia sp., Carex sp., Aster diplostephioides (DC.) C.B. Clarke, Polygonum sibiricum, Gueldenstaedtia verna, Leontopodium sp.

Note [page 5]: Carex spp.

Highlight [page 5]: Indicator species analysis showed that Carex sp.

Note [page 6]: Plural “Sedges” not “a sedge” - if it means more Carex species.

Highlight [page 6]: Belowground biomass,relative cover of sedge and forb showed a significant difference between the two zonal types of the study sites (seasonally frozen ground and permafrost zones).

Note [page 6]: Do you mean “Community cover” (= total cover of certain community on the site) or “Species cover” of particular species. It should be written clearly.

Note [page 6]: What does it mean? It should be specified, which functional traits/species decreased and which ones increased.

Highlight [page 6]: Community cover, relative covers of sedges and forbs decreased/increased substantially at the ALT of greater than…

Note [page 7]: Maybe they should be sorted alphabetically for easier orientation with the sites.

Highlight [page 7]: Sites

Note [page 7]: Why not just “East”

Highlight [page 7]: east-facing

Note [page 7]: East?

Highlight [page 7]: east-facing

Note [page 7]: Is 6 residuum from previous version?

Highlight [page 7]: 6/south

Note [page 7]: Same with 5

Highlight [page 7]: 5/south

Note [page 7]: Residuum?

Highlight [page 7]: 9/south

Note [page 9]: No need to be repeated here, it is obvious from the graphs, where it is already stated.

Highlight [page 9]: Notes: A, soil water content; B, soil pH; C, silt content; D, soil total carbon; E, soil total nitrogen and; F, soil total phosphorous.

Note [page 9]: A sentence must be finished with a full stop. I suggest to add a note, that using a particular test was based on normality of tested data.

Highlight [page 9]: b) above the bars means significant difference between sites at permafrost and seasonally frozen ground using the Student’s t tests or Mann-Whitney U tests

Note [page 10]: Repetitive information as it is in the graphs, no need to add it here again. Sure, I agree that various figures should be marked here with various letters (A to F for better reference in the text, but it is not necessary to writ explanation to the headers of the figures since it is stated on y-axis of each graph.

Highlight [page 10]: Notes: A, community cover; B, below-ground biomass; C, Species richness; D, Pielou evenness; E, sedge relative cover, and; F, forb relative cover. N=13; data were averaged for the site level.

Note [page 12]: Same as above, no need to repeat explanation of the letters A-F here.

Highlight [page 12]: Notes: A, soil water content; B, soil pH; C, silt content; D, soil total carbon; E, soil total nitrogen, and; F, soil total

phosphorous.

Highlight [page 14]: At the permafrost sites, vegetation indices significantly changed with the increase of ALT. Community cover and sedge relative cover reached the peak when the ALT was about 2.0 m while

Pielou evenness index and sedge relative cover showed the opposite trend, and below-ground

Deleted: P<0.05, *,

Deleted: Correlation between environmental factors

Deleted: -

Deleted: -

Deleted: -

Deleted: -

Deleted: -

Deleted: ¶

Formatted: Font: Not Bold

Deleted: as

Formatted: Font color: Light Blue

Deleted: [44]

Formatted: Font color: Light Blue

Deleted: [45, 46]

Deleted: c

Deleted: ,

Note [page 14]: This is a result, not a discussion, please move it above or change the text, thus you will not repeat the results here and you will start with explanation what your results mean and what are the relationships with other studies.

Highlight [page 15]: Plants 2020, 9, x FOR PEER REVIEW

15 of 21

biomass and species richness significantly (p<0.05) decreased with increasing ALT (Figure 3).

Highlight [page 15]: Meanwhile, the NMDS analysis showed more similar vegetation compositions at sites with the ALT at 0.6-2.0 m (Figure 4). However, at sites with the ALT at 2.0–3.5 m, vegetation indices changed dramatically (Figures 3 and 4). Furthermore, among all the vegetation indices, below-ground biomass relative cover of sedge and forb were significantly higher at permafrost sites than that at sites in areas of seasonally frozen ground (Table 3). Although the community cover did not show significant difference, it was lower at sites in areas of seasonally frozen ground, suggesting that the disappearance of permafrost could result in a decrease of vegetation cover as a result of the declined coverage of sedges.

Note [page 15]: These sentences described the results, it so not a discussion.

Note [page 15]: Only one unknown Carex or there is potential that more cares species were on the site?

Highlight [page 15]: Indicator species analysis showed that Carex sp.

Note [page 16]: Not an abstract, this is a conclusion, these sentence is superfluous here.

Note [page 16]: I suggest to start as: “Our results showed…”

Highlight [page 16]: The results showed that community cover, below-ground biomass, plant species richness, and relative cover of sedges decreased with deepening active layer, while Pielou evenness and forb relative cover showed a contrary trend.

Strikeout [page 16]: In this study, the relationships among ALT and vegetation features were investigated.

Author Response

Dear reviewer 3,

Thank you very much for your comments and suggestions. We appreciate your hard work. 

We revised the manuscript according to your comments. Please see the attachment.

Best regards,

Authors
